# Synthesis of Carvacrol Derivatives as Potential New Anticancer Agent against Lung Cancer

**DOI:** 10.3390/molecules27144597

**Published:** 2022-07-19

**Authors:** Anu Bansal, Md. Moshfekus Saleh-E-In, Pallab Kar, Ayan Roy, Neeta Raj Sharma

**Affiliations:** 1School of Bioengineering and Biosciences, Lovely Professional University, Phagwara 144411, India; anu.21971@lpu.co.in; 2Division of Forest Resources, College of Forest and Environmental Sciences, Kangwon National University, Chunchon 200701, Korea; saleheinm@kangwon.ac.kr; 3B.S. Diagnostic and Pathology Laboratory, Siliguri 734001, India; pallabkar.bio@gmail.com; 4Mailman School of Public Health, Columbia University, New York, NY 10032, USA; ayanroy.bio@gmail.com

**Keywords:** apoptosis, carvacrol, carvacrol aldehyde, copper–Schiff base complex, lung cancer, A549 cells

## Abstract

Lung cancer remains a major public health concern among all cancer diseases due to the toxicity and side-effects of the available commercially synthesized drugs. Natural product-derived synthesized anticancer drugs are now of promising interest to fight against cancer death. Carvacrol is a major component of most essential oil-bearing plants with potential pharmacological activity, especially against various cancer cell lines. Among the other organometallic compounds, copper complexes have been reported to be effective anticancer agents against various cancer cell lines, especially lung and leukemia cancers, due to the nontoxic nature of copper in normal cells since it is an endogenic metal. In this study, we synthesized three carvacrol derivatives, i.e., carvacrol aldehyde, Schiff base, and copper–Schiff base complex, through an established synthesis protocol and characterized the synthesized product using various spectroscopic techniques. The synthesized derivatives were evaluated for in vitro cytotoxic activity against different cancer cell lines, including human lung cancer (A549) and human fibroblast (BALB-3T3). Our findings showed that the copper–Schiff base complex derived from carvacrol inhibited the proliferation and migration of the A549 cell lines in a dose-dependent manner. This activity might be due to the inhibition of cell proliferation and migration at the G_2_/M cell-cycle phase, as well as apoptosis, possibly through the activation of the mitochondrial apoptotic pathway. To our knowledge, this is the first report on the activity of the copper–Schiff base complex of carvacrol against A549 cell lines. Our result highlights that a new synthesized copper complex from carvacrol could be a novel potential drug in the treatment of lung cancer.

## 1. Introduction

Cancer is a fatal disease which caused 10 million deaths worldwide in 2020, while lung cancer contributed to 2.21 million cases and is expected to rise by 2040 [1]. Abnormal cell growth is the main characteristic underlying cancer. It is often the result of a cascade of mutations that lead to deregulation of the receptors that control cell growth, apoptosis, and DNA repair. Surgery, radiation, chemotherapy, and hormonal and targeted therapy are various methods used to treat cancer. However, treatments have evolved with the understanding of underlying biological processes [2]. Despite the progress in the field of cancer research, the entire globe is in the grip of this life-threatening disease. Therefore, there is a need to discover alternative and efficient therapeutic drugs with no side-effects [2].

Carvacrol is a common essential oil component that belongs to the monoterpene class with a phenolic configuration. It is most abundantly found in *Origanum vulgare* (72.71%), *Thymus sipyleus* (37.95%), *Monarda didyma* (22.3%), and other essential oil-bearing plants [3,4,5]. It is a generally regarded as safe (GRAS) food flavoring compound for human consumption (also safe for mammals) and does not cause any toxicity [6]. It possesses great biological potential and is used to treat many diseases including hypertension, immune system modulators, diabetes, and cancer [7,8,9,10]. Recently, it has emerged as a prominent anticancer agent to treat different types of cancer cell lines such as prostate, colon, liver, breast, cervical, lung, and neuroblastoma (N2a). Studies have shown that carvacrol can target cancer cells by inhibiting cell proliferation, migration, and invasion via cell-cycle arrest, DNA damage, and altering the expression level of different protein markers involved in extrinsic or intrinsic apoptosis [11]. On the other hand, carvacrol can also be synthesized in the laboratory via different methods, including chemical and biotechnological synthesis through metabolic engineered microorganisms [12]. It was reported that derivatives of carvacrol, where the acidic proton (H^+^) of carvacrol was substituted by different esters, ethers, and acetic acid functional groups, demonstrated antioxidant and cytotoxic activity against HeLa cells [13].

In recent years, metal-based complexes have generated wide interest in research and in the pharmaceutical industry due to their wide range of biological and pharmaceutical properties [14]. Furthermore, many metals, especially d-block metals (transition metals), have been investigated for cytotoxic potential owing to their chemical properties [15]. The coordination of a metal with small organic molecules can modify the geometry and stereochemistry of the parent molecule, further improving its efficacy. So far, several metals including platinum, ruthenium, rhenium, copper, and osmium have been investigated as possible metals to form coordination complexes with anticancer potential. Supermolecular chemistry based on the noncovalent interactions and modifications in the chemical structure provides target-based supermolecular chemotherapy against cancer cells [16,17,18]. Although metallodrugs are used as anticancer drugs, such as cisplatin (a platinum derivative), many disadvantages have been reported with respect to their cytotoxic effects such as neuropathic, neurotoxic, and neurological dysfunctions [19]. Among all metal-based drugs, research is more focused on copper complexes to investigate their cytotoxicity due to the fact that copper is an endogenic metal and is nontoxic in normal cells in comparison to cancer cells. Copper complexes have been reported as anticancer agents against various cancer cell lines, most notably lung and leukemia cancers, by inhibiting the activity of the proteasome, transcription factors, enzymes, and protein markers of cancer cells, thereby leading to apoptosis [20,21]. The overall mechanism of action is depicted in Figure 1. In light of these facts and, in response to the current interest in herbal-based anticancer drugs, the aim of the present study was to synthesize a new class of carvacrol derivatives with copper to form a carvacrol–copper complex, and to assess the in vitro cytotoxic potential and possible mechanistic action of the synthesized products for an evaluation of their cytotoxic activity against lung cancer cells, which could lead to the future development of an anticancer drug with high therapeutic value.

## 2. Results and Discussion

### 2.1. Synthesis of Carvacrol Aldehyde (2-Hydroxy-3-methyl-6-(propan-2-yl)benzaldehyde)

In the synthesis process, acetylation and alkylation took place in the presence of SnCl_2_, Et_3_N, and paraformaldehyde ((CH_2_O)n) to maximize the yield of aldehyde product from the reaction exclusively at the *ortho*-position [22]. The *ortho*-chelation in carvacrol is provided by SnCl_2_ due to its charge density, which is further stabilized by Et_3_N. Paraformaldehyde acts as an electrophile agent in this reaction [23]. 

Carvacrol acts as a weak acid and loses its hydroxyl proton during formylation, which leads to the generation of a carvacrol anion. Thereafter, the electron density is increased on the benzene ring of carvacrol. The reaction with SnCl_2_ leads to the formation of a carvacrol–SnCl_2_ complex as an intermediate alcohol derivative. Furthermore, the reaction with paraformaldehyde causes the oxidation of the alcoholic group on the carvacrol–SnCl_2_ complex followed by the conversion of paraformaldehyde to a methyl derivative during the reduction process. The aldehyde group of the carvacrol–SnCl_2_ complex is donated by paraformaldehyde, specifically at *ortho*-position of the carvacrol. As a result, a pale-yellow liquid of carvacrol aldehyde is obtained. The carvacrol aldehyde synthesis scheme is presented in the Appendix A.

The characterization of carvacrol aldehyde was achieved with the help of ^1^H-NMR spectroscopy, mass spectrometry, and FTIR spectroscopy. To compare the structure of the pure compound with reference to the ^1^H-NMR spectrum, the presence of an aromatic compound was indicated by a sharp peak at 7.9 ppm, highlighting the substitution of the hydroxyl group in the aromatic ring. The sharp peaks at 1.5–2.2 ppm indicated the presence of a methyl group in the structure. The peak at 10.3 ppm was attributed to the presence of the aldehyde group, whereas the hydroxyl group (OH) was observed at 13.17 ppm. The molecular weight of the structure was confirmed by the ion peak at *m*/*z* 179.30 in the mass spectrum. The FTIR spectrum showed significant peaks with a stretching band corresponding to the aromatic C–H bond at ν 2965 cm^−1^, whereas the band at ν 2926 cm^−1^ corresponded to the aldehyde CH group. The carbonyl stretching frequencies were observed at ν 1627 cm^−1^. The spectroscopic data of carvacrol aldehyde are presented in the Appendix A.

### 2.2. Synthesis of Carvacrol-Derived Schiff Base

Existing studies have emphasized the development of Schiff bases derived from aromatic aldehydes *ortho*-substituted with a hydroxyl (−OH) group because they have the ability to serve as a bidentate ligand for transition metal ions [24]. In the present study, we synthesized the Schiff base from a mixture of carvacrol aldehyde and 2-aminophenol in ethanol followed by the addition of concentrated HCl at 75–80 °C. Following various steps, the Schiff base (yield: 120 mg) was recovered. The molecular weight was confirmed by the mass spectrum with a molecular ion peak at *m*/*z* 269. The FTIR spectrum of the Schiff base showed a significant peak at ν 2959 cm^−1^, corresponding to the aromatic C–H bond with stretching vibration. Moreover, the other bands corresponding to aldehydic C=N, C=C, and N–O groups were also observed at 1593 cm^−1^, 1556 cm^−1^, and 1223 cm^−1^, respectively. The spectroscopic data of carvacrol-derived Schiff base are presented in the Appendix A. In addition, the scheme (Appendix A) of carvacrol-derived Schiff base synthesis is provided in the Appendix A.

### 2.3. Synthesis of Cu(II)–Schiff Base Complex

In the present study, green crystals of the copper–Schiff base complex (100 mg) were synthesized using the Schiff base ligand derived from carvacrol (Figure 2). The surface morphology of the copper–Schiff base complex was studied through SEM and EDX spectroscopy (Appendix A). The SEM micrographs revealed that the complex possessed a uniform surface. The EDX peaks confirmed the occurrence of the copper element in the complex. The elemental composition along with relative proportions was revealed by EDX analysis. The SEM and EDX analysis confirmed the copper percentage (4.99%) and weight percentage (21.03%) in the complex. The sharp peaks of XRD patterns indicated the crystalline nature of the copper–Schiff base complex, as recorded in the range of 2θ = 2–60° (Appendix A). The copper–Schiff base complex displayed peaks at 2θ = 7°, 10°, 11°, and 18°. Similarly, the Cu(II) complex with a bidentate N_2_O_2_ Schiff base ligand showed peaks at 2θ = 10°, 11°, 17°, 19°, and 21°. Hence, our findings are in support of earlier studies, confirming the formation of the copper–Schiff base complex [25].

The zeta potential is used to measure surface charges on compounds and determine their degree of repulsion in a suspension. By virtue of the electrostatic repulsion among individual particles in suspension, a good stability can be indicated by a zeta potential <−30 mV or >+30 mV of the solution. To determine the stability of the copper–Schiff base complex, it was dispersed in ethanol at 30 °C and analyzed (Appendix A). As a result, three peaks were observed. The first peak had an average of 34.3 ± 49.3 mV, and its area contributed to 53% of the potential distribution. The second peak was at −94.4 ± 35.2 mV, with a contributed area of 33.8% of the potential distribution. The third peak was observed at 122 ± 12.9 mV, and its area contributed to 13.2% of the potential distribution. The average of all analyzed peaks was −53.5 mV. These results indicate the stability of the copper–Schiff base complex.

### 2.4. Effect of Carvacrol Derivatives on Cell Viability

The cytotoxicity in the cells can be determined by cell membrane leakage or the inhibition of cytoskeleton components such as actin filaments and microtubules. The cellular cytotoxic effects of (i) carvacrol aldehyde, (ii) the Schiff base, and (iii) the copper–Schiff base complex were evaluated using the A549 cell line, along with the BALB-3T3 cell line as a control. For quantitation of cells, the in vitro cellular viability was assessed by the MTT assay in both cell lines. The yellow-colored MTT is reduced to purple formazan in the mitochondria of living cells by the action of mitochondrial dehydrogenase. The absorbance of the solution was measured by a UV spectrophotometer at 520 nm.

The effect of different concentrations (3.9, 7.8, 15.6, 31.25, 62.5, 125, 250, and 500 µg/mL) of carvacrol aldehyde, the Schiff base, and the copper–Schiff base complex on the cell viability of A549 cells after 24 h treatment is shown in Figure 3A–C. During the exposure of carvacrol aldehyde, the Schiff base, and the copper–Schiff base complex, the treated cells reduced their growth and showed significant inhibition of cell viability, with *p* < 0.05 at the highest concentration studied (500 µg/mL) in A549 cell lines. Carvacrol aldehyde, the Schiff base, and the copper–Schiff base complex exhibited 43.6%, 53.76%, and 28.29% cell viability against the A549 cell line and 53.5%, 53.3%, and 55.3% cell viability against the BALB-3T3 cell line at a dosage of 500 µg/mL, respectively. According to ISO EN 10993, the synthesized compounds exhibited cell viability of more than 50% in the case of normal healthy cell line; therefore, they could be considered as practically noncytotoxic against healthy cell lines.

The kinetics of cell viability inhibition by all derivatives treated with A549 cells differed initially, compared to the untreated control cells. As a result of drug exposure, the cell viability inhibition was dose-dependently increased for all tested cells over 24 h. Our results suggested that all the three compounds showed a marked effect against the A549 cell line. Therefore, a reduction in cell viability of A549 cancer cells was triggered by the drug exposure as a function of the concentration. The IC_50_ values of A549 cells were found to be 278.3 ± 4.33, 492.79 ± 4.05, and 233.49 ± 4.18 µg/mL for carvacrol aldehyde, the Schiff base, and the copper–Schiff base complex, respectively. In the case of the BALB-3T3 cell line, the corresponding values were 447 ± 1.23, 488 ± 1.54, and 508 ± 1.36 µg/mL, respectively (Figure 4).

According to the current investigation, the in vitro cytotoxic effect of the synthesized compounds (carvacrol aldehyde, Schiff base, and copper–Schiff base complex) exhibited significant activity through the MTT assay. The compounds caused cell death via apoptosis by targeting biological molecules such as DNA, RNA, and their associated enzymes either directly or indirectly, as reported earlier [26]. In our study, the copper–Schiff base complex showed potent cytotoxic activity against the A549 cells with a significantly less toxic effect on the control cell line (BALB-3T3) as compared to the other two compounds, carvacrol aldehyde and the Schiff base. Moreover, the synthesized copper–Schiff base complex showed an IC_50_ value of 233.39 ± 4.18 µg/mL (66.17 µM), which is significantly lower than that of the precursor carvacrol (reported as 500 µM to 1000 µM against the A549 cell line) [27]. In the current study, the copper–Schiff base complex led to a more efficient reduction in cancer cell viability as compared to carvacrol aldehyde and the Schiff base; therefore, its cytotoxic potential and mechanistic action were subsequently analyzed.

The cytotoxicity toward cancer cells can also be determined by measuring the activity of cytoplasmic enzymes released by damaged cells, such as lactate dehydrogenase (LDH). Rupture of the cell membrane results in the release of LDH in the cytoplasm; therefore, LDH acts as a key indicator of cell apoptosis or cell damage [28]. The amount of LDH is increased in tumor cells due to their hypoxic microenvironment [29]; accordingly, they alter their metabolism by converting pyruvate into lactate via LDH in the presence of oxygen to avoid oxidative stress on the mitochondria. This is known as the Warburg effect. Tumor cells undergo this modification to increase aerobic metabolism to promote cell proliferation and growth [30]. The tumor cell damage following treatment with certain drugs that release LDH can be easily quantified by using the NADH produced during the conversion of lactate to pyruvate. As shown in Figure 5, a significant change was observed in the level of LDH when the A549 cell line was treated at varying concentrations (7.9 to 500 µg/mL) of the copper–Schiff base complex. A significant increase in the LDH release in A549 cells was observed after treatment with the copper–Schiff base complex in a dose-dependent manner. A maximum cell cytotoxicity of 78.9% was induced in A549 cells by the copper–Schiff base complex at the highest concentration of 500 µg/mL. 

A previous study showed that two novel synthesized copper complexes, namely, copper bis(4-methyl-3-thiosemicarbazonato) acenaphthenequinone and copper bis(4-ethyl-3-thiosemicarbazonato) acenaphthenequinone, displayed cytotoxic potential against the IGROV (human ovarian cancer cells) and MCF-7 (human breast adenocarcinoma cells) cell lines according to the LDH assay. The study revealed that these metal complexes had efficient cytotoxic potential with a reduced IC_50_ value as compared to cisplatin [31]. In our study, the copper–Schiff base complex induced the release of LDH by disrupting the cell membrane. The results indicate that the copper–Schiff base complex showed significant cytotoxicity against A549 cell lines.

### 2.5. Inhibition of Cell Invasion by Copper–Schiff Base Complex

Metastasis is the leading cause of cancer-related deaths, accounting for 90% of all deaths [32]. Metastasis accelerates the migration and invasion of tumor cells from the primary neoplasm to the blood, lymphatic vessels, and distant tissues or organs (secondary site), while further supporting the growth of tumor cells at the secondary site [33]. In the present work, the transwell assay was used to determine whether the copper–Schiff base complex affected the invasiveness and migration of A549 cells. The total number of A549 cells present in 100 µL of cell suspension was counted to be 5.75 × 10^4^ using a hemocytometer. Untreated cells (negative control) revealed the migration of 1.2 × 10^4^ cells in the presence of a chemo-attractant (FBS), whereas cells treated with concentrations of 125, 250, and 500 µg/mL revealed the migration of 3.3 × 10^4^, 2.57 × 10^4^, and 2.07 × 10^4^ cells, respectively.

The results showed that the highest concentration (500 µg/mL) significantly reduced the FBS-induced invasion of A549 cells when compared to the other concentrations (125 and 250 µg/mL), as shown in Figure 6. The above result suggested that the chemo-attractant increased the migration ability of the cells, but its effect could be reduced following treatment with the copper–Schiff base complex.

Similarly, the in vitro and in vivo cytotoxic activity of the disulfiram/copper (DSF/Cu) complex was reported against NSCLC (non-small-cell lung cancer) cells [34]. These studies are in support of our findings, indicating that the copper–Schiff base complex-treated A549 cells showed a significant reduction in migration and invasion in a dose-dependent manner by inhibiting matrix metalloproteinases (MMP-2 and MMP-9).

### 2.6. Cytotoxic Mechanistic Action of Copper–Schiff Base Complex 

#### 2.6.1. Apoptosis Assessment

In order to investigate the cytotoxic potential of the copper–Schiff base complex toward the A549 cancer cell line and to delineate whether the cells underwent apoptosis (cell death), we used annexin V/propidium iodide staining. In this study, annexin V alone can give a false-positive result due to necrosis. In the biochemical manifestation of apoptosis, the presence of phosphatidylserine in the plasma membrane (on the outer layer) of the cell undergoing apoptosis acts as a signal for the macrophages to eliminate apoptotic cells [35]. Annexin V labeled with fluorescein isothiocyanate (FITC) can be applied to detect the presence of phosphatidylserine via Ca^2+^-catalyzed binding using fluorescence microscopy [36].

#### 2.6.2. Terminal Deoxynucleotidyl Transferase Nick-End Labeling (TUNEL) Assay

Fragmentation of DNA is one of the main aspects of apoptosis. To study the mechanism underlying the toxicity of the copper–Schiff base complex toward A549 human lung cancer cells, a TUNEL assay was performed. TUNEL-positive cells are an indicator of cells undergoing apoptosis due to DNA fragmentation. The negative control (TdT enzyme omitted) did not show positive staining, whereas the positive control (deoxyribonuclease used to fragment DNA prior to the terminal deoxynucleotidyl transferase reaction) showed 70.54% ± 3.22% positive staining. TUNEL-positive cells (apoptotic cells) (12.64% ± 1.22%, 41.31% ± 1.26%, and 48.47% ± 1.48%) were observed when the copper complex was used to treat cells at the concentrations of 125, 250, and 500 μg/mL, respectively, as compared with the negative control (Figure 7A–D).

The highest percentage of apoptotic cells in the TUNEL assay was observed at a concentration of 500 μg/mL of the copper–Schiff base complex, as shown in Figure 7E. Jiyong and colleagues studied the interaction of a benzimidazole-based copper complex with DNA, which could cause perturbations in the DNA structure and induce DNA cleavage [37]. Similar results were obtained with the other two copper complexes, [Cu(bpbb)_0.5_**•**Cl**•**SCN]**•**(CH_3_OH) and [Cu_2_(bpbb)**•**Br_3_**•**(OH)]_n_, against the human colon cancer cell line (HCT-116). The complex [Cu(bpbb)_0.5_**•**Cl**•**SCN]**•**(CH_3_OH) was involved in more efficiently damaging the DNA of cancer cells by intercalating with DNA and generating intracellular ROS, thereby inducing cell apoptosis [38]. Likewise, our results showed that the copper–Schiff base complex induced apoptosis in a dose-dependent manner by initiating DNA fragmentation, assupported by the TUNEL assay.

### 2.7. Effect of Copper–Schiff Base Complex on Various Phases of Cell Cycle

G_1_, S, G_2_, and M are the four sequential phases of the cell cycle, and their regulation is influenced by factors such as DNA integrity, cell size, and extracellular growth signals. Cell proliferation becomes unusual when the cell cycle is deregulated. To determine the effect of the copper–Schiff base complex on the cell cycle of A549 lung cancer cells stained with propidium iodide, the complex was applied at varying concentrations (125, 250, and 500 μg/mL) for 24 h by flow cytometry (Figure 8A–D). The flow cytometric analysis was performed using annexin V/PI staining to visualize the cell cycle. Variation in the cell distribution following treatment with the copper–Schiff base complex was observed in each phase of the cell cycle (Figure 8E). The copper–Schiff base complex at various dosages (125, 250, and 500 μg/mL) decreased G_1_ cells by 11.67%, 14.67%, and 30.67% and S-phase cells by 2%, 4.33%, and 8%, respectively, when compared to the untreated cells. The complex also induced an accumulation in G_2_/M-phase cells from 12.33% (control) to 20%, 28%, and 37%, at 125, 250, and 500 μg/mL, respectively (Figure 8A–D).

In another study, cell-cycle arrest was induced by Cu(SBCM)_2_ at the G_2_/M phase, as indicated by the expression of p53 protein in MCF-7 cells [39]. Likewise, in another study, *Ruta graveolens* (Ruta) induced apoptosis via cell-cycle arrest in the COLO-205 cell line (colon cancer) at the G_2_/M phase by upregulating the expression of p21 and p27 [40]. In the current investigation, the copper–Schiff base complex also induced G_2_/M-stage cell-cycle arrest in A549 cells. The initiation of cell-cycle arrest was almost certainly caused by the damage of intracellular DNA in A549 cells, as evidenced by the TUNEL assay. This finding implies that the distribution of cells in various stages was obstructed at the G_2_/M phase when A549 cells were treated with the copper–Schiff base complex. The G_2_/M phase acts as a checkpoint, preventing DNA-damaged cells from entering mitosis, making it a potential target for cancer therapy. However, the mechanism underlying G_2_/M-stage arrest has remained elusive and needs to be further investigated.

### 2.8. Copper Complex Elicits Caspase-Dependent Apoptosis in A549 Cell Lines

Studies have revealed that the selective anticancer potential of copper is due to its reduction from the Cu^2+^ to Cu^+^ state in the anoxic environment of cancer cells [41]. In tumor cells, Cu^+^ acts as a catalyst for reactive oxygen and nitrogen species generation, thereby inducing apoptosis. Intrinsic or mitochondria-mediated apoptosis is induced by the retreat of growth factors and generation of intracellular mitochondrial stress, whereas dysregulation in transmembrane death receptors such as Fas-receptor (FADD) initiates the extrinsic apoptosis pathway. The regulation of intrinsic or extrinsic apoptosis is based on the B-cell lymphoma 2 (Bcl-2) (antiapoptotic) and caspase families of proteins [42,43]. In the case of the intrinsic apoptosis pathway, Bcl-2 associated X protein (Bax) is activated, followed by initiation of the mitochondrial outer membrane permeabilization (MOMP), which further results in the discharge of cytochrome-c (proapoptotic protein) from the intermembrane space into the cytosol [44,45,46]. The released cytochrome c helps in the binding of apaf-1 (apoptotic protease activating factor 1) and inactive caspase-9 to form the apoptosome. Thereafter, caspase-9 transforms into its active state and further activates effector caspases such as caspase-3 or caspase-7.

Consequently, to understand the mode of the apoptotic pathway (intrinsic or extrinsic) activated by the copper–Schiff base complex after providing treatment in the A549 cell lines, the mRNA transcription levels of antiapoptotic gene Bcl-2 and proapoptotic factors caspase-3, caspase-9, and Bax were evaluated using RT-PCR and the relative quantification PCR technique.

According to the results, the expression of Bcl-2 was reduced after 48 h, while the expression level of Bax was simultaneously increased by the copper–Schiff base complex in a dose-dependent manner (Figure 9A). Subsequently, we analyzed the effect of the novel synthesized copper–Schiff base complex on the mRNA expression levels of caspase-3 and -9, familiar apoptotic proteins. A significant increase was observed in the mRNA levels of caspase-3 and caspase-9 after 48 h treatment with the copper–Schiff base complex (Figure 9B). These data support the perception that the novel copper–Schiff base complex could induce apoptotic alterations inside A549 cells by altering Bax and Bcl-2 levels and upregulating caspase-3 and caspase-9 mRNA levels.

It has been validated that increased manifestation of Bcl-2 family proteins (antiapoptotic) inhibits apoptosis, thus representing a specific target in cancer therapeutics [47]. Proteins of the Bcl-2 family are key regulators of the intrinsic (mitochondrial) apoptotic pathway [48]. The Bcl-2 protein initiates cytochrome c release from the inner mitochondria and causes cell death via apoptosis. A similar result with an increased level of Bcl-2 expression in leukemia cells (drug-resistant) was reported by Sakamoto et al. [49]. To verify the in-depth role of the copper–Schiff base complex in inducing mitochondrial apoptosis, the expression of Bcl-2 (antiapoptotic) and Bax (proapoptotic), as well as of caspase-3 and caspase-9, was analyzed by qRT-PCR. A previous study by Cabrera and colleagues revealed that, after treating leukemia cells with benzophenone thiosemicarbazone, the Bcl-2 expression was inhibited via a reduction in Bcl-2 phosphorylation and G_2_/M cell-cycle arrest [48]. Our findings demonstrated that G_2_/M cell-cycle arrest was induced by the novel copper–Schiff base complex in a dose-dependent manner with an increase in Bax mRNA expression and a concomitant reduction in Bcl-2 mRNA expression as compared to the untreated control cells. As observed through various studies, the balance between Bax and Bcl-2 is a key element for cell survival or cell death [47]. Treatment with the copper–Schiff base complex was found to enhance the Bax/Bcl-2 ratio in A549 cells after 48 h, resulting in an elevated level of apoptosis. Simultaneously, we analyzed the effect of the novel copper–Schiff base complex on the mRNA level of caspase-3 and -9 in the A549 cells. Caspase-9 (a cysteine-aspartic protease) is an initiator caspase that, once activated, transforms downstream effector caspases such as caspase-3 into their active form, which are in turn responsible for the initiation of intrinsic apoptosis by promoting apoptosis via DNA fragmentation and inhibiting the DNA repair mechanism [50].

## 3. Materials and Methods

### 3.1. Chemicals and Reagents

In the present study, all reagents and chemicals were of AR grade and procured from Sigma-Aldrich.

### 3.2. Synthesis of Carvacrol Aldehyde (2-Hydroxy-3-methyl-6-(propan-2-yl)benzaldehyde)

First, 8.1 g of triethylamine (Et_3_N) and 4.8 g of anhydrous stannous chloride (SnCl_2_) were mixed with 210 mL of toluene. The reaction mixture was stirred continuously for 20 min, followed by the addition of 30 g of carvacrol under reflux at 50–60 °C. Thereafter, the intermediate (organic layer) was produced and mixed with paraformaldehyde (13.8 g), followed by continuously refluxing at 50–60 °C for 2–3 h [22].

### 3.3. Synthesis of Carvacrol-Derived Schiff Base

Carvacrol aldehyde (100 mg) and 2-aminophenol (60 mg) were dissolved in 5 mL of ethanol and stirred for 15 min, followed by dropwise addition of concentrated HCl. Gradually, the temperature was increased to 80 °C. The aqueous layer was separated and washed with 10 mL of demineralized water. Subsequently, ethyl acetate (10 mL) was added and washed with brine solution (10 mL). The compound was filtered through sodium sulfate (Na_2_SO_4_) (10 gm), and the precipitate was vacuumed at a temperature below 50 °C. Finally, the pure orange Schiff base (yield: 120 mg) was recovered and collected [51].

### 3.4. Synthesis of Cu(II)–Schiff Base Complex

The Schiff base ligand (200 mg) and copper acetate (50 mg) were added to an ethanol solution (10 mL) and continuously stirred at 25–30 °C for 10–15 min. The mixture was further heated to 80–85 °C for the next 6 h. The mixture was gradually cooled down at room temperature (25 °C), which produced a green precipitate product. Then, the precipitate was filtered and washed with precooled ethanol for further analysis [52].

### 3.5. In Vitro Studies

The experimental cell lines, A549 (human lung cancer) and BALB-3T3 (human fibroblast), were obtained from NCCS, Pune. Cryo-preserved cells (contained in the vials) were sprayed with 70% ethanol solution and placed in a sterile laminar chamber with airflow. The content was transferred into a presterilized centrifuge tube containing culture medium (2 mL) and centrifuged at 1200 rpm at room temperature for 10 min. After discarding the supernatant, the remaining pellets were washed using phosphate-buffered saline (PBS) to removing any remaining dimethyl sulfoxide (DMSO). The pellets were again resuspended in 1 mL of Dulbecco’s modified Eagle medium (DMEM) and Ham’s F12k medium. Subsequently, the experimental cells were incubated at 37 ± 1 °C in a 5% CO_2_ incubator chamber supplemented with >90% humidity for 24 h. The cells were examined in terms of their morphology (round, plump, or elongated), attachment, growth, and confluency after 24 h under a microscope. Then, the cells were refed with a fresh culture medium (5 mL) and incubated until 70–80% confluency.

### 3.6. Detection of Cell Viability Using MTT Assay

Carvacrol aldehyde, the Schiff base, and the copper–Schiff base complex were weighed and dissolved in DMSO to attain the desired concentrations through serial dilution with DMSO (500 to 0.039 µg/mL) at room temperature. A549 and BALB-3T3 cells were exposed to 100 mL of the test compounds, and then incubated at 37 ± 1 °C in a 5% CO_2_ incubator chamber supplemented with >90% humidity for 24 h, along with the positive control (500 µg/mL SDS), negative control, and blank cells. After 24 h treatment, the cells were examined under a phase-contrast microscope to check for morphological changes. The culture medium was removed, and MTT solution (50 mL, 1 mg/mL) was added to each well. The plates were further incubated, and 100 mL of isopropanol was added to each well after applying the MTT solution, followed by complete dissolution of the crystals in order to record the absorbance at 570 nm [53]. Untreated cells were considered as 100% viable.

### 3.7. Lactate Dehydrogenase (LDH) Release Assay

The LDH assay was employed for detecting the cell cytotoxicity. In this experiment, A549 cells were placed in a 96-well plate, followed by treatment with the copper–Schiff base complex for 24 h at varying concentrations ranging from 500 to 725 µg/mL. LDH is a soluble cytosolic enzyme found in the majority of eukaryotic cells. As a result of cell death, owing to the damage of the plasma membrane, it is released into the culture medium. LDH detection was performed using a ScienCell™ LDH Cytotoxicity Assay kit (TMS, Selangor Darul Ehsan, Malaysia).

### 3.8. Transwell Invasion Assay

Serial dilution of the copper–Schiff base complex was conducted at room temperature using DMSO to attain the desired concentrations (500 to 125 µg/mL). After achieving 70–80% confluency, enzymatic digestion (Trypsin/EDTA) was performed to remove cells from the culture flasks, followed by centrifugation for 10 min (at 1200 rpm) and resuspension of cells (at a density of 1–3 × 10^5^ cells/mL) in the culture medium. Cells were added to 96-well plates and treated with the copper–Schiff base complex at the concentrations of 500, 250, and 125 µg/mL. Subsequently, cells were incubated following the standard protocol described by Justus et al. [54].

### 3.9. Analysis of Apoptosis Using TUNEL Assay

A549 cells were treated with the copper–Schiff base complex (500, 250, and 125 µg/mL) and incubated for 24 h as per a standard procedure [55], along with the positive control. The TUNEL-positive cell percentage was scored at 400× magnification under a microscope. Three independent measurements (*n* = 3) were performed in random fields.

### 3.10. Detection of Cell-Cycle Arrest Using Flow Cytometry

#### 3.10.1. Copper–Schiff Base Complex Exposure on A549 Cell Line

The copper–Schiff base complex (100 µL) (125, 250, and 500 µg/mL), positive control (500 µg/mL), negative control, and blank samples were added to a 96-well plate. Thereafter, the cells were incubated in a 5% CO_2_ incubator supplemented with >90% humidity for 24 h.

#### 3.10.2. Cell-Cycle Analysis

After the treatment, the cells were harvested, washed, and fixed overnight in ethanol (70%) at −20 °C. The cells were pelleted, washed in chilled PBS, and resuspended in staining solution (50 μg/mL). Following treatment, cells were harvested and stained using an annexin V/FITC and propidium iodide (PI) staining kit (BD Biosciences), before incubating in the dark at room temperature (20–25 °C) for 15 min. Following staining, cells were analyzed by flow cytometry. A quadrant gate was applied (according to the manufacture’s protocol) to visualize all of the dot plots corresponding to the tests, and the percentages of cells in different quadrants were calculated.

### 3.11. Reverse Transcriptase Polymerase Chain Reaction (RT-PCR) and Quantification PCR (qRT-PCR)

#### 3.11.1. RNA Separation and RNA Purification

Copper–Schiff base complex-treated A549 cell lines were separated using centrifugation. The supernatant was removed, and the cell suspension (pellet) was taken in a 15 mL tube for the isolation of RNA. The RNA was purified using the kit, followed by a guanidine/thiocyanate extraction protocol [56]. Then, the RNA was treated with RNase-free DNase I (2 units) for 30 min at 37 °C to remove genomic DNA. Finally, 10 pg of RNA was taken and resuspended in water.

#### 3.11.2. Reverse Transcription for Bcl-2, Bax, Caspase-3, and Caspase-9

The RNA sample was centrifuged momentarily for collection in the tube and then reverse-transcribed into cDNA using a random primer (0.5 μg), ImProm-II reverse transcriptase (5 units), dNTPs (0.5 mM), and RNase inhibitor (20 units). The obtained cDNA was resuspended in 10 μL of deionized water, mixed gently, and stored at −20 °C. The PCR primer sets of Bcl-2, Bax, caspase-9, and caspase-3 contained 1 nM of each primer (sense and antisense).

#### 3.11.3. Polymerase Chain Reaction (PCR) Analysis

The cDNA (2μL) was analyzed using PCR, containing 0.2 mM dNTPs, 0.4 μM of each primer (Bcl-2, Bax, caspase-9, and caspase-3), and 0.1 unit of Taq polymerase (Appendix A). In PCR analysis, GAPDH mRNA (housekeeping gene) was used to standardize and normalize the expression of target genes. The Bcl-2, Bax, caspase-3, and caspase-9 genes were amplified and analyzed. All PCR protocols, along with primer sequences, are listed in Appendix A. All amplifications were subjected to initial heating to 95 °C (2 min), followed by a final extension at the temperature of 72 °C (7 min).

#### 3.11.4. Agarose Gel Electrophoresis (AGE)

PCR products were electrophoresed in agarose/TAE gels, and each PCR product (5 μL) was analyzed. To separate the variant bands properly, the PCR products of Bax, Bcl-2, caspase-3, and caspase-9 were loaded onto a 2% agarose gel and run at 50 V/cm for 15 min. The produced DNA bands were examined in a UV transilluminator with ethidium bromide.

#### 3.11.5. Real-Time Quantitative PCR (qRT-PCR)

The qRT-PCR assay was performed with 15 μL of reaction mixture, containing 1.5 μL of single-stranded cDNA, 7.5 μL of PCR biosystems, diluted in a ratio of 1:10, 6 pmol/μL of each specific primer (Bcl-2, Bax, caspase-3, and caspase-9), and 4.8 μL of water. Reproducibility was checked by performing the PCR assay in triplicate for each sample concentration under investigation. The PCR environment was maintained as mentioned in Appendix A.

### 3.12. Statistical Analysis

The data were analyzed using GraphPad Prism Software (version 5.0). All experiments were repeated thrice, and the values were expressed as the mean ± SD for each test with the significance set at *p* < 0.05.

## 4. Conclusions

In summary, we showed that a novel carvacrol-derived Schiff base complex with copper was exceptionally active with an IC_50_ value of 233.39 ± 4.18 µg/mL (66.6 μM) as compared to carvacrol. The viability of A549 cells was reduced by this compound in a dose-dependent fashion, which promoted apoptosis through DNA fragmentation. Additionally, the carvacrol-derived Schiff base complex also instigated cell-cycle arrest at the G_2_/M phase and prompted apoptosis via the downregulation of Bcl-2 and upregulation of Bax, caspase-3, and caspase-9. The presence of the copper ion (Cu^2+^) significantly enhanced the cytotoxic activity by targeting the delivery of the ligand to the cancer cells. Considering our findings, the copper–Schiff base complex can be a promising candidate as an efficient anticancer agent for the management of lung cancer. Therefore, in vivo studies are recommended to explore the cytotoxic potential and toxic effects of the compound.

## Figures and Tables

**Figure 1 molecules-27-04597-f001:**
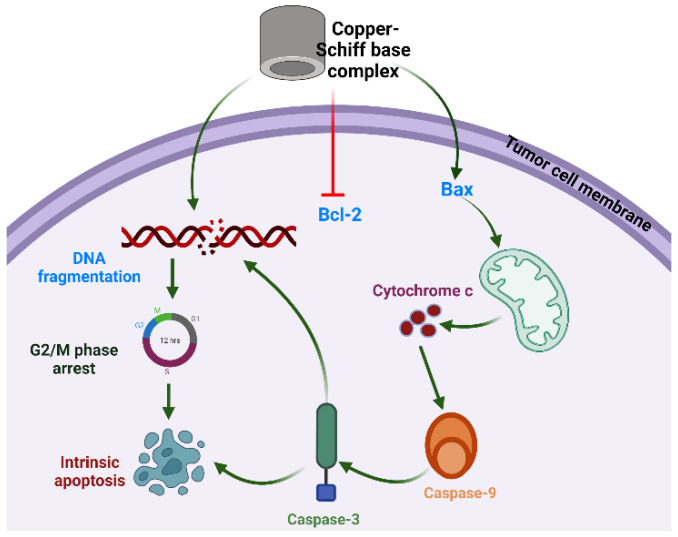
Proposed mechanism of action of copper–Schiff base complex inducing G_2_/M-phase cell-cycle arrest and intrinsic apoptotic cell death through Bax/Bcl-2 pathway in A549 cells.

**Figure 2 molecules-27-04597-f002:**
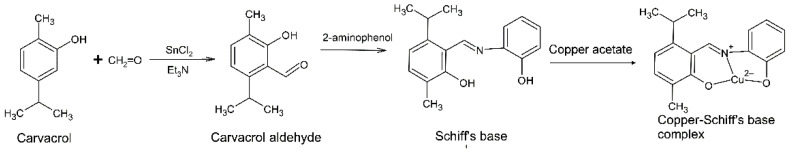
Reaction showing the synthesis of the copper–Schiff base complex from carvacrol.

**Figure 3 molecules-27-04597-f003:**
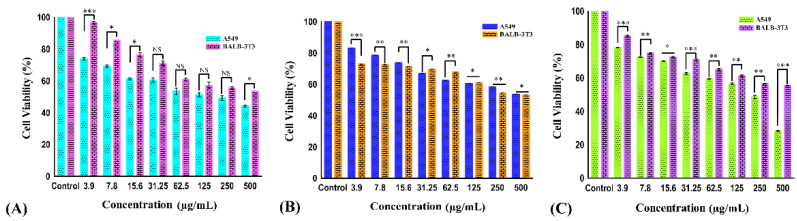
The cell viability of A549 and BALB-3T3 cell lines treated with carvacrol aldehyde (**A**), the Schiff base (**B**), and the copper–Schiff base complex (**C**) at different concentrations for 24 h. Data are expressed as the mean ± SD (*n* = 3). * *p* < 0.05; ** *p* < 0.01; *** *p* < 0.001; NS—nonsignificant.

**Figure 4 molecules-27-04597-f004:**
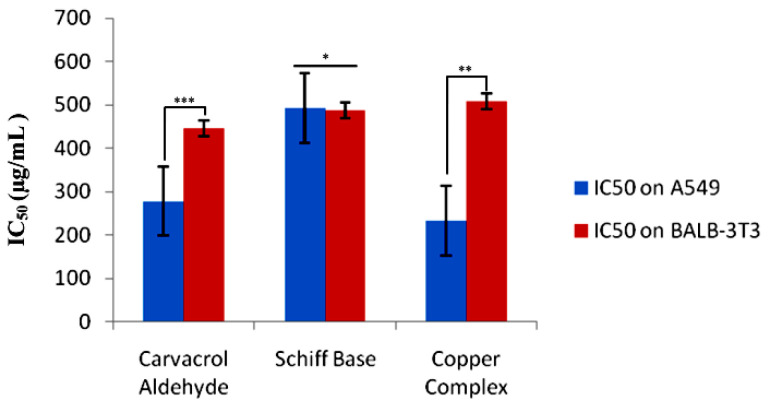
The IC_50_ values (mean ± SE) of carvacrol aldehyde, the Schiff base, and the copper–Schiff base complex in A549 and BALB-3T3 cells after 24 h of exposure. Data are expressed as the mean ± SD (*n* = 3). * *p* < 0.05; ** *p* < 0.01; *** *p* < 0.001.

**Figure 5 molecules-27-04597-f005:**
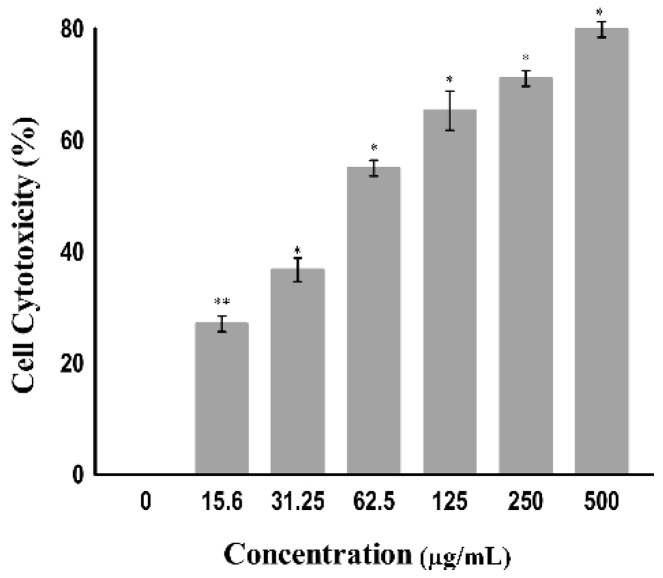
The cytotoxicity of the copper–Schiff base complex toward A549 cells using the LDH assay according to concentration. Data are expressed as the mean ± SD (*n* = 3). * *p* < 0.05; ** *p* < 0.01.

**Figure 6 molecules-27-04597-f006:**
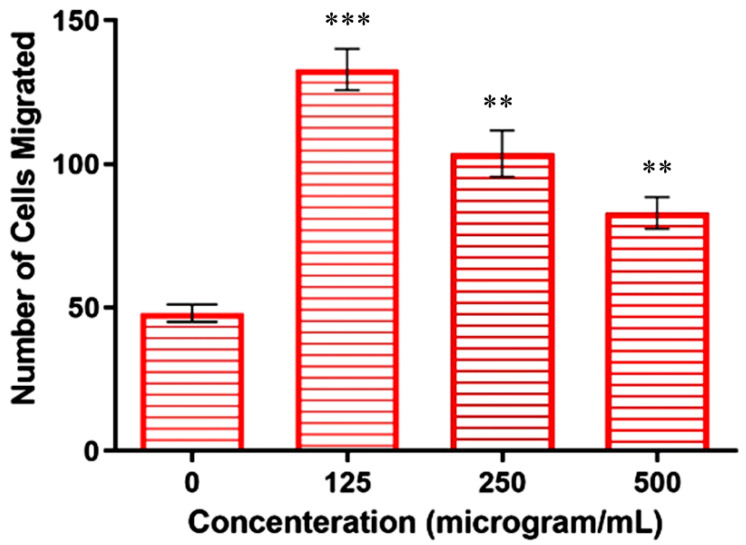
Copper–Schiff base complex inhibition of cell (A549) migration according to concentration as compared to control. Data are expressed as the mean ± SD (*n* = 3). ** *p* < 0.01; *** *p* < 0.001 when compared to control.

**Figure 7 molecules-27-04597-f007:**
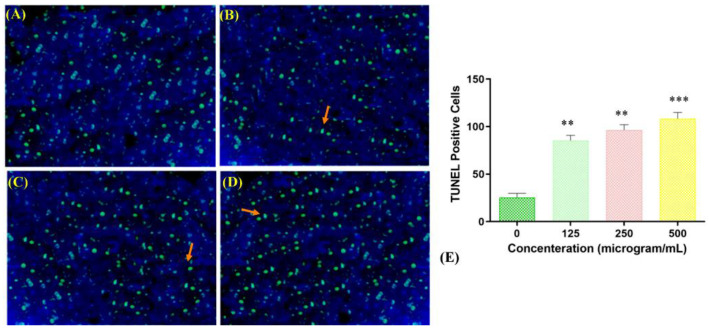
TUNEL-positive apoptotic A549 cells stained green, as indicated by arrows, for the negative control and different dosages of the copper–Schiff base complex exposed for 24 h. Ethidium bromide (C_21_H_20_BrN_3_)-stained micrograph image of DMSO-treated cells (negative control) showing the regular growth of cells (**A**); copper complex-treated A549 cells treated with dosages (µg/mL) of 125 (**B**), 250 (**C**), and 500 (**D**). A significant increase in TUNEL-positive apoptotic A549 cells was observed when treated with the 500 µg/mL dosage of the copper–Schiff base complex. (**E**) The TUNEL-positive cells were treated with the copper–Schiff base complex at different concentrations compared with the negative control. Data are expressed as the mean ± SD (*n* = 3). ** *p* < 0.01; *** *p* < 0.001 when compared to control.

**Figure 8 molecules-27-04597-f008:**
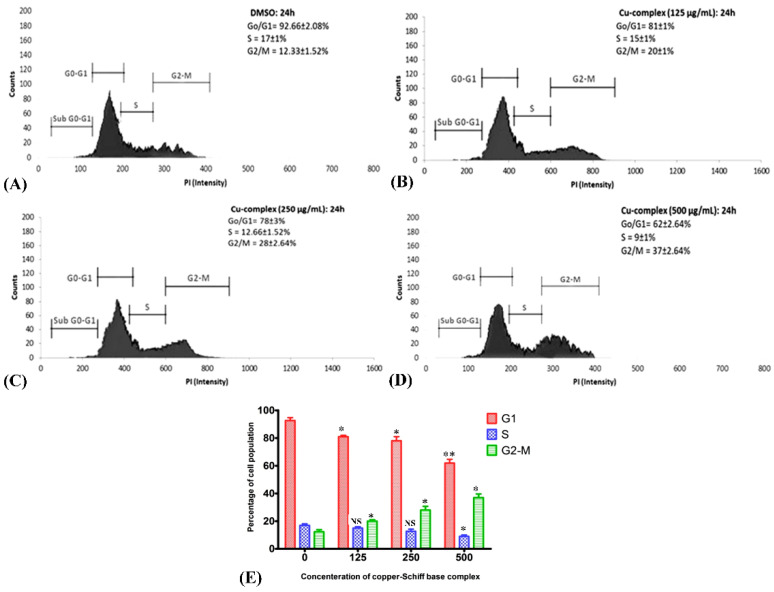
Flow cytometry cell-cycle analysis of A549 cells treated with the copper–Schiff base complex at 125, 250, and 500 μg/mL for 24 h. Histograms show the patterns of A549 cell-cycle distribution (G_0_/G_1_, S, and G_2_/M): (**A**) control, (**B**) 125 μg/mL, (**C**) 250 μg/mL, and (**D**) 500 μg/mL for 24 h. Results show the accumulation of the copper–Schiff base complex-treated cells in the G_2_/M stage. (**E**) Effect of copper–Schiff base complex (125, 250, and 500 μg/mL) on cell-cycle profile of A549 cells. Data are expressed as the mean ± SD (*n* = 3). * *p* < 0.05; ** *p* < 0.01; NS—nonsignificant when compared to control.

**Figure 9 molecules-27-04597-f009:**
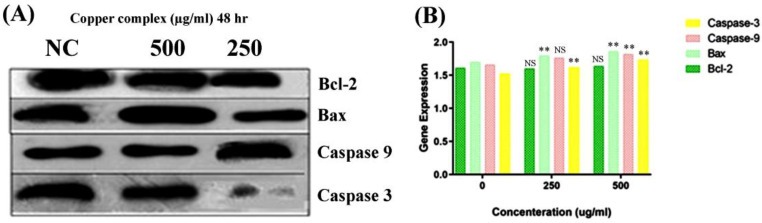
Overall expression (**A**) and relative quantification PCR analysis (**B**) of Bcl-2, Bax, caspase-3, and caspase-9 in response to the copper–Schiff base complex in A549 cells. NC: untreated control. Dose-dependent study on the global expression of Bcl-2, Bax, caspase-3, and caspase-9. The graph shows the fluorescence intensities of the PCR products versus the dose concentration. Data are expressed as the mean ± SD (*n* = 3). ** *p* < 0.01; NS—nonsignificant when compared to control.

## Data Availability

All relevant data are provided within the manuscript.

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
