# Peer review of "Synthesis of Carvacrol Derivatives as Potential New Anticancer Agent against Lung Cancer"

_molecules, 2022, doi:10.3390/molecules27144597_

Round 1

Reviewer 1 Report

I strongly recommend the manuscript for publication. It only needs a minor revision. In Figure 4, please enter the IC50 unit on the vertical axis. The figure should constitute a separate whole. At present, you need to look for a unit in the text surrounding the figure. Please also introduce the cross-references to supplementary materials  file in the Results and Discussion section, as well as in Materials and methods, e.g. concerning the structure of Schiff's base and its complex with copper.

Author Response

Reply to Reviewer’s Comments pertaining to the Manuscript Number: molecules-1813309 entitled ‘Synthesis of Carvacrol Derivatives as Potential New Anticancer Agent Against Lung Cancer’

 Dear Editors,

We highly appreciate the vital comments, suggestions and valuable time provided by the reviewers, aimed at improving the standards of the present work. We have responded meticulously to all the comments. The manuscript has been modified accordingly and is being submitted for your final consideration. We hope that the incorporated amendments comply with the reviewers’ suggestions. Responses to reviewers’ comments have been provided below.

Reviewer #1

  1. Comment: In Figure 4, please enter the IC50 unit on the vertical axis. The figure should constitute a separate whole. At present, you need to look for a unit in the text surrounding the figure.

Reply: We have incorporated the IC50 unit in the vertical axis of Figure 4 and also we have indicated the IC50 value in the respective text.

  1. Comment: Please also introduce the cross-references to supplementary materials file in the Results and Discussion section, as well as in Materials and methods, e.g., concerning the structure of Schiff's base and its complex with copper.

Reply: The references and cross references have been incorporated in the manuscript and supplementary file respectively as per recommended by the reviewer.

Reviewer 2 Report

Bansal et al describe the effects of carvacrol aldehyde, Schiff base and copper-shiff base complex, that they synthetized, on the human lung cancer cell line A549 and on the mouse fibroblast cell line BALB-3T3, showing better activity of the last one on cell viability. They also showed results of apoptotic cel death induction, effect on cell migration and cell cycle of the copper-schiff base complex.

In my opinion revisions are necessary. In particular:

INTRODUCTION:

-           some other reference should be added when they write about the use of carvacron for many diseases (lane 56-63)

-          Reference 11 is not so recent as stated (lane 68-70)

RESULTS AND DISCUSSION:

2.4

-          BALB-3T3 has been used as control; however, this is a mouse cell line. Why they did not use a human fibroplast cell line to compare it to A549 cells?

-          At 500 mg/ml also in the control cells a strong effect on cell viability is reported (about 50%). This result need to be better discussed

-          Figure 3: please, use the same scale in the y axes of the three graphs

-          Statistical analysis and variability (SD or SE) are missing in the figure 3 and in the text (lane 209-213)

-          Statistical analysis is missing in Figure 4

-          Lane 219-221: it is not clear at which compound they are referred

2.5

- LDH assay is generally used to evaluate the effects on cell viability, as the MTT assay, so it is not a surprise that the graph in figure 5 show similar results that graph in figure 4. The authors should represent the two results in the same way (histograms of viable cells or curve of cell death cells)

2.4 and 2.5 should be putted in the same paragraph

2.6

- Figure 6 and related text seem to show conflicting results: an increase in the number of migrated cells following treatment is reported in the figure, but in the textit seems that they say that something different (lane 274-282).

- statistical analysis is missing in Figure 6

2.7

Tunel assay evaluate apoptotic cells. It should be move in the follow paragraph

2.8

- the title of this paragraph is “Effect of Copper-shifft base on cell cycle, but 2.8.1 concern apoptosi assessement (put here 2.7). In addition, results of annexinV/PI experiment (I suppose that this test has been used to evaluate apoptotic cell death) shold be reported. A figure with the dot plot of AN-V/PI should be added to show results obtained.

- statistical analysis is missing in figure 7

2.9

- viability assays have been performed following 24h treatment, but PCR experiments have been performed following 48h treatment, why?

- statistical significance is reported in figure 9B; however, in the caption is indicated that each lane of the figure is representing an individual sample…

- Maybe the concentrations in figure 9A are inverted.

METHODS

-          Some methods need to be better described (or references added)

-          3.10.2: they described only PI staining method, but not the annexin/PI one.

-          3.12: the statistical analysis performed should be indicated.

Author Response

Reply to Reviewer’s Comments pertaining to the Manuscript Number: molecules-1813309 entitled ‘Synthesis of Carvacrol Derivatives as Potential New Anticancer Agent Against Lung Cancer’

Dear Editors,

We highly appreciate the vital comments, suggestions and valuable time provided by the reviewers, aimed at improving the standards of the present work. We have responded meticulously to all the comments. The manuscript has been modified accordingly and is being submitted for your final consideration. We hope that the incorporated amendments comply with the reviewers’ suggestions. Responses to reviewers’ comments have been provided below.

Reviewer #2

INTRODUCTION:

  1. Comment: some other reference should be added when they write about the use of carvacrol for many diseases (lane 56-63)

Reply: We have incorporated other references as per reviewer’s suggestion.

  1. Comment: Reference 11 is not so recent as stated (lane 68-70)

Reply: We have updated the recent reference as per reviewer’s suggestion.

RESULTS AND DISCUSSION:

  1. Comment: BALB-3T3 has been used as control; however, this is a mouse cell line. Why they did not use a human fibroplast cell line to compare it to A549 cells?

Reply: We have followed the methods as per Ivask et al., (2015) and Titma et al., (2016) with slight modifications. In these studies the authors used A549 and BALB-3T3 cell lines to check the cytotoxic potential of metal complexes. We have followed similar approach in the present study. However, as aptly suggested by the reviewer, we will include human fibroplast cell line to compare it to A549 cells in our future studies.

  1. Comment: At 500 mg/ml also in the control cells a strong effect on cell viability is reported (about 50%). This result need to be better discussed

Reply: The result and discussion have been updated as per recommendation and highlighted in the manuscript.

  1. Comment: Figure 3: please, use the same scale in the y axes of the three graphs

Reply: Y axis scale has been corrected as per suggestion.

  1. Comment: Statistical analysis and variability (SD or SE) are missing in the figure 3 and in the text (lane 209-213)

Reply: We have incorpored the statistical analysis and variability (SD or SE) in the figure 3 and in the text as per reviewer’s suggestion.

  1. Comment: Statistical analysis is missing in Figure 4.

Reply: It has been incorporated in Figure 4.

  1. Comment: Lane 219-221: it is not clear at which compound they are referred.

Reply: Sentences have been rephrased for easy understanding.

  1. Comment: LDH assay is generally used to evaluate the effects on cell viability, as the MTT assay, so it is not a surprise that the graph in figure 5 show similar results that graph in figure 4. The authors should represent the two results in the same way (histograms of viable cells or curve of cell death cells)

Reply: The suggestion has been followed and updated accordingly with the highlight in the manuscript.

  1. Comment: 2.4 and 2.5 should be putted in the same paragraph.

Reply: MTT and LDH assay have been mentioned under the same section 2.4.

  1. Comment: Figure 6 and related text seem to show conflicting results: an increase in the number of migrated cells following treatment is reported in the figure, but in the text it seems that they say that something different (lane 274-282).

Reply: The sentences have been rephrased for clear understanding which highlighted in the text.

  1. Comment: statistical analysis is missing in Figure 6.

Reply: It has been incorporated in Figure 6.

  1. Comment: Tunel assay evaluate apoptotic cells. It should be move in the follow paragraph

Reply: Changes have been done as per reviewer suggestion and highlighted in the manuscript.

  1. Comment: the title of this paragraph is “Effect of Copper-shifft base on cell cycle, but 2.8.1 concern apoptosiassessement (put here 2.7). In addition, results of annexinV/PI experiment (I suppose that this test has been used to evaluate apoptotic cell death) shold be reported. A figure with the dot plot of AN-V/PI should be added to show results obtained.

Reply: Changes have been done as suggested by the reviewer.

  1. Comment: statistical analysis is missing in figure 7

Reply: It has been incorporated in Figure 7.

  1. Comment: viability assays have been performed following 24h treatment, but PCR experiments have been performed following 48h treatment, why?

Reply: PCR study was conducted to understand the apoptotic mechanism of novel synthesized copper-Schiff complex at the gene expression level, considering the treatment effect to be more pronounced at 48h as compared to 24h treatment. Accordingly, A549 cell lines were exposed (48h), separated by centrifugation, and were allowed it settle for some time. The supernatant was removed and cell suspension was taken in tarson tube (15 mL) for the isolation of the RNA. The RNA was purified with the help of the guanidine/thiocyanate extraction method.

  1. Comment: statistical significance is reported in figure 9B; however, in the caption is indicated that each lane of the figure is representing an individual sample…

Reply: We have corrected the Figure 9B.

  1. Comment: May be the concentrations in figure 9A are inverted.

Reply: Our presented figure 9A is correct. We have introduced the PCR product of NC, 500 then 250 (µg/ml) (concentration higher to lower) as per our standard protocol.

METHODS:

  1. Comment: Some methods need to be better described (or references added)

Reply: Methodology section has been revised thoroughly with the inclusion of relevant references.

  1. Comment: 3.10.2: they described only PI staining method, but not the annexin/PI one.

Reply: Methodology section has been revised and updated as suggested by the reviewer.

  1. Comment: 3.12: the statistical analysis performed should be indicated.

Reply: We have indicated the statistical analysis in the respective section.

Reviewer 3 Report

Comments to the Authors:

This manuscript describes a novel carvacrol derived Schiff base complex with copper would be a novel potential drug in the treatment of lung cancer by inhibiting the proliferation and migration, down-regulation of Bcl-2 as well as up-regulation of Bax. The synthesized derivatives were evaluated for in vitro cytotoxic activity against different cancer cell lines using MTT, LDH release assay and so on. The copper-Schiff based complex induced apoptosis by initiating DNA fragmentation. The conclusions were verified by enough and convincing data. It will be a solid contribution to Molecules and will certainly appeal to many of its readers. I address some of the main issues with the manuscript in the next few paragraphs. It is recommended that this manuscript to be published in Molecules after completing minor revision.

1. Study of carvacrol derivatives on cell viability after 24 h, how about 12 h and 36 h? It should be completed to prove whether time has an effect on it.

2. Figure 5, the maximum cell cytotoxicity at 500 µg/mL, but it's still growing, how about 600 µg/mL, 700 µg/mL?

3. Cell invasion inhibition by Copper-Schiff base complex, adding cytotoxic activity in vivo.

4. Introduction, the authors talked about the treatments have evolved with the understanding of underlying biological processes, supramolecular strategies should be mentioned. In order to support this statement, the following recently published important related papers should be cited: Chem. Soc. Rev. 2017, 46, 7021; Chem. Soc. Rev. 2021, 50, 2839.; Adv.Mater., 2022, 34, 2106388.

Author Response

Reply to Reviewer’s Comments pertaining to the Manuscript Number: molecules-1813309 entitled ‘Synthesis of Carvacrol Derivatives as Potential New Anticancer Agent Against Lung Cancer’

Dear Editors,

We highly appreciate the vital comments, suggestions and valuable time provided by the reviewers, aimed at improving the standards of the present work. We have responded meticulously to all the comments. The manuscript has been modified accordingly and is being submitted for your final consideration. We hope that the incorporated amendments comply with the reviewers’ suggestions. Responses to reviewers’ comments have been provided below.

Reviewer #3

  1. Comment: Study of carvacrol derivatives on cell viability after 24 h, how about 12 h and 36 h? It should be completed to prove whether time has an effect on it.

Reply: The minimum time required for viability test is 24 h. The newly synthesized carvacrol derivatives namely, carvacrol aldehyde, Schiff base and copper-Schiff base complex showed their cytotoxic potency at 24 h. Out of all, copper-Schiff base complex showed more than 70% cancer cell death after 24 h treatment in a dose dependent manner. Therefore, duration of treatment was not extended in present study.

  1. Comment: Figure 5, the maximum cell cytotoxicity at 500 µg/mL, but it's still growing, how about 600 µg/mL, 700 µg/mL?

Reply: The result obtained with the highest concentration of 500 µg/mL was helpful in elucidating the cytotoxic effect of novel synthesized copper-Schiff base complex in in vitro condition. The concentration more than 500 µg/mL might cause the cytotoxicity against the healthy cells.

  1. Comment: Cell invasion inhibition by Copper-Schiff base complex, adding cytotoxic activity in vivo.

Reply: Copper-Schiff base complex inhibits in vitro cancer cell invasion in a dose dependent manner; however, its effect was not studied in vivo in present study.

  1. Comment: Introduction, the authors talked about the treatments have evolved with the understanding of underlying biological processes, supramolecular strategies should be mentioned. In order to support this statement, the following recently published important related papers should be cited: Chem. Soc. Rev. 2017, 46, 7021; Chem. Soc. Rev. 2021, 50, 2839.; Adv.Mater., 2022, 34, 2106388.

Reply: The suggestion and the citations have been added in the introduction part of the manuscript.

Round 2

Reviewer 2 Report

Most of the question have been answered. However, some minor revisions are still needed:

- number of experiments need to be indicated in the figure legends

- the authors used AnnexinV/PI labelling to evaluate cell cycle and it is a questionable choise. As a matter of fact, PI staining of DNA require cell fixation to allow total DNA staining, while AnnV/PI staining is generally used to discriminate necrotic/late apoptotic cells from viable/early apoptotic cells and it doesn't require cell fixation. It is still not clear why they did it.

- LDH viability assay seems does not show differences in the effect of carvacrol aldehyde and copper complex on A549 cells. Did the authors make a comparsion (and evaluate statistical differences)? 

Author Response

Reply to Reviewer’s Comments pertaining to the Manuscript Number: molecules-1813309 entitled ‘Synthesis of Carvacrol Derivatives as Potential New Anticancer Agent Against Lung Cancer’

Dear Editors,

We highly appreciate the vital comments, suggestions and valuable time provided by the reviewers, aimed at improving the standards of the present work. We have responded meticulously to all the comments. The manuscript has been modified accordingly and is being submitted for your final consideration. We hope that the incorporated amendments comply with the reviewers’ suggestions. Responses to reviewers’ comments have been provided below.

Reviewer #2

  1. Comment: Number of experiments need to be indicated in the figure legends

Reply: We have incorporated number of experiments in each figure legends as per reviewer’s suggestion.

  1. Comment: The authors used AnnexinV/PI labelling to evaluate cell cycle and it is a questionable choise. As a matter of fact, PI staining of DNA require cell fixation to allow total DNA staining, while AnnV/PI staining is generally used to discriminate necrotic/late apoptotic cells from viable/early apoptotic cells and it doesn't require cell fixation. It is still not clear why they did it.

Reply: During experiment, annexinV was added in cells, and incubated for 15 minutes, afterwards PI stain was added and incubated for 15 minutes. Further, A549 cells were resuspended in 70% ethanol (fixative) solution and stored overnight and analyzed using flow cytometry to determine the number of cells in each phase of cell cycle. AnnexinV/PI staining with fixation improves the assessment of cell death. Annexin V conjugated with PI helps in determining apoptosis occurred due to DNA damage.

  1. Comment: LDH viability assay seems does not show differences in the effect of carvacrol aldehyde and copper complex on A549 cells. Did the authors make a comparsion (and evaluate statistical differences)?

Reply: Cytotoxic effect of carvacrol aldehyde, Schiff base and copper-Schiff base complex on A549 was observed in MTT assay. Based on the MTT result, copper-Schiff base complex was selected for further analysis using LDH assay in which its effect was studied on A549 cell lines by varying concentration.  We have incorporated the statistical values in Figure 5 as per reviewer’s suggestion.